Whole mitochondrial genome sequencing of Malaysian patients with cardiomyopathy

http://orcid.org/0000-0001-6196-5250 Kuan Sheh Wen 1
http://orcid.org/0000-0002-0452-9994 Chua Kek Heng 1
http://orcid.org/0000-0001-8764-130X Tan E-Wei 1
Tan Lay Koon 2
http://orcid.org/0000-0001-8810-9948 Loch Alexander 3
http://orcid.org/0000-0003-1528-3990 Kee Boon Pin 1 bpkee@um.edu.my
1 Department of Biomedical Science, Faculty of Medicine, University of Malaya , Kuala Lumpur , Malaysia
2 National Heart Institute , Kuala Lumpur , Malaysia
3 Department of Medicine, Faculty of Medicine, University of Malaya , Kuala Lumpur , Malaysia
Adnan Mohd
Electronic publication date: 2022 Apr 14
Publication date: 2022
Volume: 10
Electronic Location ID: e13265
Received 2021 Nov 9; Accepted 2022 Mar 23
Copyright: © 2022 Kuan et al.
Copyright year: 2022
Copyright holder: Kuan et al.
License: This is an open access article distributed under the terms of the Creative Commons Attribution License, which permits unrestricted use, distribution, reproduction and adaptation in any medium and for any purpose provided that it is properly attributed. For attribution, the original author(s), title, publication source (PeerJ) and either DOI or URL of the article must be cited.
License URL: https://creativecommons.org/licenses/by/4.0/

Keywords: Cardiomyopathy, HCM, DCM, Whole mitochondrial genome sequencing, mtDNA, NGS

Funding: University of Malaya GPF005C-2018 Frontier Research Fund 2017 FG014/17AFR Ministry of Higher Education Malaysia FP088/2019A and FRGS/1/2019/SKK08/UM/02/14 This work was supported by the University of Malaya Faculty Research Grant (GPF005C-2018), the Frontier Research Fund 2017 (FG014/17AFR), and the Fundamental Research Grant Scheme; Ministry of Higher Education Malaysia (FP088/2019A; FRGS/1/2019/SKK08/UM/02/14). The funders had no role in study design, data collection and analysis, decision to publish, or preparation of the manuscript.

==============================
Cardiomyopathy (CMP) constitutes a diverse group of myocardium diseases affecting the pumping ability of the heart. Genetic predisposition is among the major factors affecting the development of CMP. Globally, there are over 100 genes in autosomal and mitochondrial DNA (mtDNA) that have been reported to be associated with the pathogenesis of CMP. However, most of the genetic studies have been conducted in Western countries, with limited data being available for the Asian population. Therefore, this study aims to investigate the mutation spectrum in the mitochondrial genome of 145 CMP patients in Malaysia. Long-range PCR was employed to amplify the entire mtDNA, and whole mitochondrial genome sequencing was conducted on the MiSeq platform. Raw data was quality checked, mapped, and aligned to the revised Cambridge Reference Sequence (rCRS). Variants were named, annotated, and filtered. The sequencing revealed 1,077 variants, including 18 novel and 17 CMP and/or mitochondrial disease-associated variants after filtering. In-silico predictions suggested that three of the novel variants (m.8573G>C, m.11916T>A and m.11918T>G) in this study are potentially pathogenic. Two confirmed pathogenic variants (m.1555A>G and m.11778G>A) were also found in the CMP patients. The findings of this study shed light on the distribution of mitochondrial mutations in Malaysian CMP patients. Further functional studies are required to elucidate the role of these variants in the development of CMP.

Introduction

Cardiomyopathy (CMP) is commonly known to affect the heart muscle and its ability to pump blood around the body. In the early stages of CMP, patients are commonly asymptomatic. However, with disease progression, symptoms of heart failure such as exertional dyspnoea, fatigue and pedal oedema are often presented. CMP is one of the common causes of heart failure (Dokainish et al., 2017). There are several subtypes of CMP, including hypertrophic cardiomyopathy (HCM), dilated cardiomyopathy (DCM), restrictive cardiomyopathy (RCM), and left ventricular non-compaction cardiomyopathy (LVNC) (Elliott et al., 2008). HCM and DCM are the two most common subtypes of CMP. HCM is characterised by left ventricular hypertrophy due to cardiomyocyte hypertrophy, whereas DCM is characterised by enlarged cardiac chambers and systolic dysfunction. The estimated prevalence of HCM in the general population was 1 in 500 to 200 individuals (Semsarian et al., 2015), while the prevalence of DCM was 1 in 250 individuals (Hershberger, Hedges & Morales, 2013). In fact, the prevalence may be higher as the estimations made were probably conservative (McKenna & Judge, 2021). Nevertheless, the exact aetiology of CMP remains inconclusive. It is generally accepted that environmental and genetic factors play vital roles in the CMP onset. Approximately 20–60% of CMP cases were reported as familial (Marian & Braunwald, 2017; Sweet, Taylor & Mestroni, 2015). Mitochondrial inheritance was known to contribute to the onset of CMP.

Mitochondria are the powerhouses of cells because of their ability to generate adenosine triphosphate (ATP) molecules through oxidative phosphorylation (OXPHOS) carried out by electron transport chain (ETC) complexes. Mitochondria are responsible for cell death regulation by releasing cytochrome c to initiate necrosis and apoptosis under stress conditions (Bock & Tait, 2020). Cardiomyocytes are among the high energy demand cells necessitated by the significant workload of blood pumping. Mitochondrial dysfunction can result in inadequate ATP production, leaving the energy demand of cardiomyocytes unsatisfied and ultimately leading to CMP (Ramaccini et al., 2020) or the worst, heart failure (Chistiakov et al., 2018). The mitochondrion is one of the sites where reactive oxygen species (ROS) are generated as a by-product of OXPHOS. Defective mitochondria also increase ROS production, triggering the opening of mitochondrial permeability transition pores and stimulating the release of cytochrome c that activates cardiac cell death (Zorov et al., 2000). The destruction of cardiomyocytes can cause CMP by deteriorating heart pumping ability.

Mitochondrial DNA (mtDNA) is a small circular double-stranded DNA with 16,569 bp that can only be transmitted maternally. It contains 37 genes that encode two ribosomal RNAs (rRNAs), 22 transfer RNAs (tRNAs), and 13 protein complexes that are involved in ETC (Andrews et al., 1999). The only non-coding region in mtDNA is the D-loop that is located at positions 16,024 to 576, serving as the origin of the transcription and promoter region.

Mutations in mtDNA were reported in both HCM and DCM patients. The m.3243A>G mutation in MT-TL1 gene is among the most common mitochondrial tRNA mutations reported in CMP (Hollingsworth et al., 2012). Studies have reported m.1555A>G mutation in the MT-RNR1 gene which encodes 12S rRNA in a patient with maternally inherited CMP (Santorelli et al., 1999). Mutations in protein-coding genes such as MT-ND1 and MT-CYB may directly impact the OXPHOS as these genes are vital to encode subunits of complexes involved in ETC. The m.3395A>G mutation in MT-ND1 gene (Alila et al., 2016) and m.15482T>C mutation in MT-CYB (Hagen et al., 2013a) were reported in HCM and DCM patients, respectively. However, mitochondrial haplogroups were also frequently reported to be associated with CMP. A study in Spain reported that individuals with haplogroup H had a higher risk of developing DCM (Fernandez-Caggiano et al., 2013). Another study conducted in Denmark identified haplogroups H and HV as the risk factors for HCM development. In contrast, individuals with haplogroups J and UK had a lower risk to develop HCM (Hagen et al., 2013b).

Most studies on mitochondrial mutations in CMP patients were carried out in Western countries, rarely in Asia, and none in Malaysia. Therefore, this study is important as the first study on the full mutation spectrum of the mitochondrial genome in Malaysian CMP patients. The findings in the present study are irrefutably fundamental to fill the gap of knowledge for the mtDNA spectrum in Malaysian CMP patients.

Materials and Methods

Subject recruitment

A total of 145 CMP patients were recruited from the University of Malaya Medical Centre and the National Heart Institute Malaysia. Among these patients, 87 were diagnosed with DCM whereas 58 were diagnosed with HCM. Only patients who met the diagnostic criteria for DCM (Mathew et al., 2017) and HCM (Pantazis et al., 2015) were included in this study. For DCM patients, the inclusion criterion was ejection fraction ≤35%. Patients with the following causes of reduced ejection fraction were excluded: (1) tachycardia-induced CMP, (2) peripartum CMP, (3) ischaemic CMP in the presence of significant coronary artery disease or a history of ischemic events, (4) toxin-induced CMP, (5) ethanol-induced CMP, (6) hypertension-induced CMP, and (7) CMP due to valvular or congenital heart disease. As for HCM, the diagnosis was confirmed when patients fulfilled the American Heart Association’s echocardiographic criteria for HCM in the absence of severe uncontrolled hypertension or aortic valve diseases. Written informed consent was obtained from all participants in this study before venous blood collection. This study was conducted in accordance with the ethical principles in the Declaration of Helsinki. Ethical approvals were obtained from the Medical Ethics Committee of the University of Malaya Medical Centre (MECID NO.: 20152-1016) and the Ethics Committee of the National Heart Institute of Malaysia [IJNEC NO.: IJNEC/08/2015(3)]. Demographic and clinical data were retrieved from hospital databases.

Whole mitochondrial genome sequencing

A conventional phenol-chloroform extraction method was employed for genomic DNA extraction from the venous blood (Chua et al., 2009). The quality and quantity of the extracted genomic DNA samples were assessed via NanoPhotometer® (Implen GmbH, Munich, Germany). Long-range polymerase chain reaction (PCR) was performed with PrimeSTAR® GXL DNA Polymerase (Takara Bio, Shiga, Japan) to amplify two fragments (9.1 kb and 11.2 kb) that cover the entire mitochondrial genome, with two overlapping regions. Three sets of primer pairs: (1) MTL-F1 (5′-AAAGCACATACCAAGGCCAC-3′), MTL-R1 (5′-TTGGCTCTCCTTGCAAAGTT-3′); (2) MTL-F2 (5′-TATCCGCCATCCCATACATT-3′), MTL-R2 (5′-AATGTTGAGCCGTAGATGCC-3′) (Stawski, 2013) and (3) MTL-F3 (5′-CTTACTTCTCTTCCTTCTCTCCTTA-3′), MTL-R3 (5′-TAGTGAGGAAAGTTGAGCCAAT-3′) were used for the amplification of the mitochondrial genome. The third primer pairs were used for samples that did not amplify well with the second set of primer pairs. For PCR conditions, three-step PCR was performed to amplify both fragments: 30 cycles of denaturation at 98 °C for 10 s, annealing at 60 °C for 15 s, and extension at 68 °C for 90 s and 120 s for 9.1 kb and 11.2 kb fragments, respectively. To ascertain the amplicon size, all the PCR products were electrophoresed on 1% (w/v) agarose gel. Next, the amplicons were normalised and pooled prior to clean-up via QIAquick PCR Purification Kit (QIAGEN, Hilden, Germany). The purified amplicons were quantified using Qubit™ 1X dsDNA HS Assay Kit (Thermo Fisher Scientific, Waltham, MA, USA) with Qubit 4 Fluorometer (Thermo Fisher Scientific, Waltham, MA, USA). A total of 250 ng of pooled amplicon for each sample was input for individual library preparation. According to the manufacturer’s protocol, libraries were prepared and indexed with Nextera DNA Flex Library Prep Kit and Nextera™ DNA CD Indexes (Illumina, San Diego, CA, USA). For fragment size determination and quality checking, the prepared libraries were examined with Agilent 2100 Bioanalyzer instrument (Agilent Technologies, Santa Clara, CA, USA) via Agilent High Sensitivity DNA Kit. The prepared libraries were then quantified using Qubit™ 1X dsDNA HS Assay Kit with Qubit 4 Fluorometer. All the libraries were normalised to a single 4 nM final library for sequencing. Sequencing was performed in Illumina MiSeq system with Illumina MiSeq Reagent Kit v2 for paired-end (2 × 150 bp) reads. Illumina PhiX Control was included as an internal control for sequencing quality monitoring.

Sequence analysis

FASTQ files were input to Illumina DRAGEN FastQC + MultiQC for initial quality checking of the sequencing data. FastQC generated the overview of quality parameters such as input reads quality score, GC content, sequence length distribution, and duplication levels, while MultiQC allowed summary and visualisation of results across all samples. The reference used was Hg38 Alt-Aware, with HLAs. Next, the FASTQ files were input to Illumina mtDNA Variant Processor. Adapters were trimmed and reads with short amplicon were removed. BWA-MEM algorithm in Burrows–Wheeler Alignment Tool was used to align and map the sequence read to the revised Cambridge Reference Sequence (rCRS) (Li, 2013). Only reads with Q-score >30, analysis threshold >10%, interpretation threshold >25% and read count >10 were included for variant calling. The resulting VCF files were then input into Illumina mtDNA Variant Analyzer for visualisation of coverage and variant points position in comparison to rCRS. Variants were annotated with BaseSpace Variant Interpreter (v2.11), and Universal mtDNA Variant Converter and One Stop Annotation (mvTool v.6) in MSeqDR (Shen et al., 2018). The variants were compared and cross-checked with reliable databases that are still being maintained and constantly updated such as MITOMAP, Genome Aggregation Database (gnomAD), HmtDB, HmtVar, MSeqDR and ClinVar for their clinical significance. MITOMAP and gnomAD were referred for the allele frequencies of variants in various populations. Haplogroup assignment was conducted with HaploGrep 2 to determine the haplogroup of each subject based on their mitochondrial sequence (Weissensteiner et al., 2016). Circa (http://omgenomics.com/circa/) was used to generate the visualisation of variants distribution across the whole mitochondrial genome. The overview of the whole mitochondrial genome sequencing and analysis in this study is shown in Fig. 1.

Figure 1 Overview of whole mitochondrial genome sequencing in Malaysian CMP patients.

The workflow demonstrates the summary of steps involved from sample collection, quality check of raw sequencing data, sequence alignment, variant calling, variant annotation, variant filtering, and lastly the final identification of novel variants and CMP and/or mitochondrial diseases-associated variants with potential pathogenicity.

Conservation index (CI) analysis

MITOMASTER was utilised to evaluate the evolutionary conservation of the variants across different species. The following 15 species were selected for the calculation of CI: (1) Homo sapiens, (2) Cebus albifrons, (3) Gorilla gorilla, (4) Pan troglodytes, (5) Pongo pygmaeus, (6) Bos taurus, (7) Canis lupus familiaris, (8) Balaenoptera musculus, (9) Rhinoceros unicornis, (10) Mus musculus, (11) Macropus robustus, (12) Cyprinus carpio, (13) Drosophila melanogaster, (14) Gallus gallus and (15) Xenopus laevis. The variant with CI >75% was considered highly conserved and was expected to have functional potential. Besides, Mamit-tRNA was used to assess the conservation of variants located in tRNA across 34 species in the Euarchontoglires clade (Putz et al., 2007). The CI was listed as the number of times in which the wild-type nucleotide appears per total of 34 species.

In-silico analysis

In-silico analysis was carried out to unravel the effect of non-synonymous substitution on the structure and function of the protein. Prediction tools such as PolyPhen-2, SIFT, PROVEAN, MutPred, Mitoclass.1, CADD, FatHmm, PANTHER, SNPs&GO and PhD-SNP were used to predict the impact of the variants on the protein function (Adzhubei et al., 2010; Calabrese et al., 2009; Capriotti, Calabrese & Casadio, 2006; Choi & Chan, 2015; Kircher et al., 2014; Li et al., 2009; Martin-Navarro et al., 2017; Mi, Muruganujan & Thomas, 2013; Shihab et al., 2013; Sim et al., 2012). Meta-predictors such as MToolBox in MseqDR (Calabrese et al., 2014) and APOGEE in MitImpact (Castellana et al., 2017) were also used to assess the pathogenicity of mitochondrial genome variants. For tRNA variants, MitoTIP in MITOMAP was used to predict the disease-causing likelihood of tRNA coding variant according to their variant history, position, conservation, and secondary structure score (Sonney et al., 2017). The scoring and classification of pathogenicity of each prediction tool is shown in Table S1.

Statistical analysis

Statistical analysis was carried out via tools available from http://www.biostathandbook.com/fishers.html. Fisher’s exact test was conducted for association analysis based on the mutation frequency and clinical data. The significance level of all tests was set at P value < 0.05.

Results

Demographic and clinical data of CMP patients

Among the 145 patients, 95 (65.5%) were male and 50 (34.5%) were female. The age of the patients ranged from 19 to 82 years old and the average age was 51.20 ± 15.54 years. The majority of the patients comprised of three of the main ethnic groups in Malaysia, i.e., 64 (44.1%) Malay, 55 (37.9%) Chinese, 19 (13.1%) Indian, and seven (4.8%) were of other ethnicities. Among the CMP patients, 87 (60.0%) were DCM patients while 58 (40.0%) were HCM patients. The demographic and echocardiographic data are shown in Table 1.

Table 1 Demographic and echocardiographic data of CMP patients in Malaysia.

Demographic data	Frequency, n (%)	
		DCM (N = 87)	HCM (N = 58)	
Gender	Male	60 (69.0)	35 (60.3)	
Female	27 (31.0)	23 (39.7)	
Ethnicity	Malay	44 (50.6)	20 (34.5)	
Chinese	31 (35.6)	24 (41.4)	
Indian	9 (10.3)	10 (17.2)	
Other	3 (3.4)	4 (6.9)	
Age		48.40 ± 15.64	55.44 ± 14.39	
Echocardiographic parameters			
IVSd (cm ± SD)	1.01 ± 0.27	1.69 ± 0.52	
LVPWd (cm ± SD)	1.00 ± 0.26	1.21 ± 0.36	
LVIDd (cm ± SD)	6.37 ± 0.87	4.55 ± 1.11	
LVIDs (cm ± SD)	5.44 ± 1.13	2.82 ± 1.28	
LA diameter (cm ± SD)	4.26 ± 0.85	4.02 ± 0.86	
Ejection fraction (% ± SD)	27.21 ± 14.59	65.57 ± 15.86	
Note:

IVSd, Interventricular septal thickness at diastole; LVPWd, Left ventricular posterior wall thickness at diastole; LVIDd, Left ventricular internal diameter at end diastole; LVIDs, Left ventricular internal diameter at end systole; LA diameter, Left atrial diameter.

Whole mitochondrial genome sequencing analysis

Overall, 99.64% of sequence reads were mapped to the reference. An average of 94.55% of reads had a mapping quality (MAPQ) score ≥40. For the minimum base call quality (Q30), an average of 87.60% of total bases achieved ≥Q30. The average coverage of the whole mitochondrial genome for all the samples was 1,140.8 ± 703.8×.

Multi-step filtering procedure as illustrated in Fig. 1 was carried out on all the observed mitochondrial variants. In the first step, variants with read depth <10× and variant read frequency <0.05 were removed from further analysis. As a result, the whole mitochondrial genome sequencing revealed 1,077 variants among the CMP patients, with an average of 40.7 variants detected per patient. Subsequently, 565 variants that met the following conditions were excluded: (1) Variants fulfilled BA1 and/or BS1 criteria in American College of Medical Genetics and Genomics and the Association for Molecular Pathology (ACMG/AMP) 2015 guidelines (Richards et al., 2015), (2) Variants that were markers found at ≥80% in haplogroups (letter-number-letter level), and (3) Variants in the control region that may not have clinical significance (Tables S2–S6). The remaining 512 variants consisted of 308 (60.2%) synonymous, 18 (3.5%) novel, 36 (7.0%) rRNA, 29 (5.7%) tRNA, 119 (23.3%) non-synonymous, one (0.2%) start loss and one (0.2%) stop-retained variation.

Novel variants

Variants that were not reported in MITOMAP and/or gnomAD were considered as novel variants in this study. A total of 18 novel variants were detected in the CMP patients (Table 2; Fig. 2), including 10 (55.6%) non-synonymous, six (33.3%) in tRNA and two (11.1%) in rRNA. The non-synonymous variants were: two heteroplasmic variants in MT-ND1 (m.3511A>C and m.3577A>C), one heteroplasmic variant in MT-CO1 (m.7416T>C), one heteroplasmic variant in MT-ATP6 (m.8573G>C), one homoplasmic variant in MT-CO3 (m.9318C>A), one homoplasmic variant in MT-ND3 (m.10144G>T), two homoplasmic variants in MT-ND4 (m.11916T>A and m.11918T>G) and two homoplasmic variants in MT-ND5 (m.12595A>G and m.12663C>A). For rRNA, one homoplasmic variant was found in MT-RNR1 (m.1008C>T), while the remaining five variants were found heteroplasmic in MT-RNR2 (m.1731A>G, m.2465T>A, m.2473A>C, m.2475T>C, m.2479C>A). The variants found in tRNA were m.4313T>A in MT-TI and m.5551C>T in MT-TW; both in the homoplasmic state. The in-silico analysis of pathogenicity for the novel variants is illustrated in a heatmap (Fig. 3; Table S7). Interestingly, three variants (m.8573G>C, m.11916T>A and m.11918T>G) were predicted to have a strong negative impact on the protein function by most of the in-silico predictors; thus, their effects on the onset of CMP could not be neglected.

Figure 2 Circos plot showing the distribution of variants across the whole mitochondrial genome.

The outer ring shows the gene positions, followed by the specific location of novel variants (pink ring) and CMP and/or mitochondrial disease-associated variants (yellow ring).

Figure 3 Heatmap of conservation index (CI) and pathogenicity of variants predicted by in-silico tools.

The legend of each predictor is shown as a colour scale at the right, with the minimum and maximum score labelled on top of each scale. Only Mitoclass.1 has no score but category. Variant with no result predicted by the in-silico tool is labelled as N/A. Note that the in-silico tools are not able to predict the pathogenicity of non-protein-coding rRNA and tRNA variants, except MitoTIP for tRNA variant prediction.

Table 2 List of novel mitochondrial variants detected in Malaysian CMP patients.

Variant	Gene	Complex	AA change	CMP patients		
				Frequency, n (%)	Haplogroup	
				DCM (N = 87)	HCM (N = 58)	Total (N = 145)		
m.1108C>T	MT-RNR1	–	–	1 (1.1)	0 (0.0)	1 (0.7)	M12a2	
m.1731A>G	MT-RNR2	–	–	0 (0.0)	1 (1.7)	1 (0.7)	D4	
m.2465T>A	MT-RNR2	–	–	3 (3.4)	1 (1.7)	4 (2.8)	R8a1, M*, U1a1c1d	
m.2473A>C	MT-RNR2	–	–	0 (0.0)	1 (1.7)	1 (0.7)	F1f	
m.2475T>C	MT-RNR2	–	–	1 (1.1)	0 (0.0)	1 (0.7)	M71a1a	
m.2479C>A	MT-RNR2	–	–	1 (1.1)	0 (0.0)	1 (0.7)	F1e3	
m.3511A>C	MT-ND1	I	Thr → Pro	1 (1.1)	0 (0.0)	1 (0.7)	B4c1b2a2	
m.3577A>C	MT-ND1	I	Met → Leu	1 (1.1)	0 (0.0)	1 (0.7)	M18	
m.4313T>A	MT-TI	–	–	1 (1.1)	0 (0.0)	1 (0.7)	D5b4	
m.5551C>T	MT-TW	–	–	1 (1.1)	0 (0.0)	1 (0.7)	F1a3b	
m.7416T>C	MT-CO1	IV	Phe → Leu	1 (1.1)	0 (0.0)	1 (0.7)	B5a1a	
m.8573G>C	MT-ATP6	V	Gly → Ala	3 (3.4)	2 (3.4)	5 (3.4)	M8a2, B5a1	
m.9318C>A	MT-CO3	IV	His → Asn	1 (1.1)	0 (0.0)	1 (0.7)	M3a1	
m.10144G>T	MT-ND3	I	Gly → Val	1 (1.1)	0 (0.0)	1 (0.7)	M53	
m.11916T>A	MT-ND4	I	Phe → Tyr	1 (1.1)	0 (0.0)	1 (0.7)	R30b2a	
m.11918T>G	MT-ND4	I	Ser → Ala	1 (1.1)	0 (0.0)	1 (0.7)	F2b1	
m.12595A>G	MT-ND5	I	Met → Val	0 (0.0)	1 (1.7)	1 (0.7)	D4h3	
m.12663C>A	MT-ND5	I	Asn → Lys	0 (0.0)	1 (1.7)	1 (0.7)	B4a1a	
Note:

AA, amino acid; variant(s) in bold is highly potential to be pathogenic.

CMPs and/or mitochondrial disease-associated variants

After excluding synonymous and novel variants, the remaining 186 variants were subjected to filtering based on literature review and in-depth analysis to identify variants associated with CMPs and/or mitochondrial diseases. The inclusion criteria were: (1) The variant that was reported to be associated with CMPs, (2) The variant that was reported in mitochondrial diseases and had negative functional consequences predicted by at least three software and meta-predictors. As a result, 156 variants were eliminated due to the presence of functional or in-silico evidence as benign variants, or not being reported to associate with mitochondrial or any disease. After multi-step filtering procedure, as illustrated in Fig. 1, 30 variants were reported to be associated with CMPs and/or mitochondrial diseases. A total of 40 (27.6%) among 145 patients harboured these variants, whereby 20 (50.0%) of them were DCM patients and 20 (50.0%) were HCM patients.

These variants consisted of 23 non-synonymous, one start loss (m.12338T>C), one stop retained (m.14149C>T), four MT-rRNA and one MT-tRNA variations. Four (13.3%) variants were observed each in MT-ND1 and MT-ND4, three (10.0%) each in MT-RNR2, MT-ND2, MT-CO3 and MT-ND5, two (6.7%) each in MT-ND6 and MT-CYB, and one (3.3%) each in MT-RNR1, MT-TI, MT-CO1, MT-ATP8, MT-ATP6 and MT-ND3. All the variants were detected to be homoplasmic (>99%), except m.9804G>A that was heteroplasmic with 21.4% load. Among these variants, 25 variants were located in protein-coding genes involved in the respiratory chain, of which 17 (68.0%) in Complex I, two (8.0%) each in Complex III and Complex V, and four (16.0%) in Complex IV. Furthermore, two variants (m.1555A>G and m.11778G>A) were well-known confirmed pathogenic variants reported in numerous studies for their clinical significance in diseases.

Of these 30 variants, 13 variants that belonged to the mtDNA haplogroup of the proband were sorted out (Tables S8 and S9), resulting in 17 remaining variants (Table 3; Fig. 2). The status of the variants in databases and their associated diseases are listed in Table 4. Most of these variants were predicted to be potentially pathogenic by in-silico analysis (Fig. 3; Table S10).

Table 3 List of CMP and/or mitochondrial disease-associated variants with potential pathogenicity detected in Malaysian CMP patients.

Variant	Gene	Complex	AA change	MITOMAP AF	gnomAD AF	Haplogroup variant	CMP patients		
							Frequency, n (%)	Haplogroup	
							DCM (N = 87)	HCM (N = 58)	Total (N = 145)		
m.1555A>G	MT-RNR1	–	–	0.0002	0.0011	–	0 (0.0)	1 (1.7)	1 (0.7)	E1a2	
m.2905A>G	MT-RNR2	–	–	0.0004	0.0004	C7a1d	1 (1.1)	0 (0.0)	1 (0.7)	B5a1b1	
m.3202T>C	MT-RNR2	–	–	0.0006	0.0023	A2k	0 (0.0)	1 (1.7)	1 (0.7)	F2d	
m.3397A>G	MT-ND1	I	Met → Val	0.0030	0.0015	C5b1a, HV17, M27c, R0a2j, U5b2a4	2 (2.3)	0 (0.0)	2 (1.4)	M26, H13a2a	
m.4316A>G	MT-TI	–	–	0.0007	0.0004	–	0 (0.0)	1 (1.7)	1 (0.7)	M17c	
m.6340C>T	MT-CO1	IV	Thr → Ile	0.0016	0.0010	–	0 (0.0)	3 (5.2)	3 (2.1)	M7c2, B5a1b1	
m.9025G>A	MT-ATP6	V	Gly → Ser	0.0006	0.0008	–	1 (1.1)	0 (0.0)	1 (0.7)	B5a1c	
m.9214A>C	MT-CO3	IV	His → Pro	0.0002	0.0000	–	0 (0.0)	1 (1.7)	1 (0.7)	M13c	
m.9804G>A	MT-CO3	IV	Ala → Thr	0.0030	0.0036	H5a1g1, H6c	1 (1.1)	0 (0.0)	1 (0.7)	M59	
m.9856T>C	MT-CO3	IV	Ile → Thr	0.0003	0.0003	–	0 (0.0)	1 (1.7)	1 (0.7)	R9b1a3	
m.10237T>C	MT-ND3	I	Ile → Thr	0.0016	0.0018	H6a1a4, H90, J2a1a1a1, L0d1a1c	1 (1.1)	0 (0.0)	1 (0.7)	M7c1c3	
m.11084A>G	MT-ND4	I	Thr → Ala	0.0040	0.0017	A15a, M7a1a, M2a1a1a, M28a2, M52a1b1, J1b4a, C1b13a1, H1bk	0 (0.0)	1 (1.7)	1 (0.7)	B4c1b2a2	
m.11087T>C	MT-ND4	I	Phe → Leu	0.0019	0.0010	H1j1b, J1c5f, L0d1b2b1a	0 (0.0)	1 (1.7)	1 (0.7)	M81	
m.11361T>C	MT-ND4	I	Met → Thr	0.0004	0.0002	–	1 (1.1)	0 (0.0)	1 (0.7)	M12a1b	
m.11778G>A	MT-ND4	I	Arg → His	0.0036	0.0002	T3, X2p1	1 (1.1)	0 (0.0)	1 (0.7)	M59	
m.14420C>T	MT-ND6	I	Gly → Glu	0.0002	0.0023	H1h2, U5b1c1a1	0 (0.0)	1 (1.7)	1 (0.7)	G1c2	
m.14894T>C	MT-CYB	III	Phe → Leu	0.0002	0.0003	L2a3	0 (0.0)	1 (1.7)	1 (0.7)	U2c1	
Note:

AA, amino acid; AF, allele frequency; variant(s) in bold is confirmed pathogenic.

Table 4 Overview of CMP and/or mitochondrial disease-associated variants with potential pathogenicity.

Variant	MITOMAP	HmtDB	HmtVar	MSeqDR	ClinVar	Disease association	
m.1555A>G	Confirmed	Pending classification	–	Drug response	Drug response	CMP (Santorelli et al., 1999); DEAF (Maeda et al., 2020); LS (Habbane et al., 2020)	
m.2905A>G	Polymorphism	Pending classification	–	–	–	LVNC (Ross et al., 2020)	
m.3202T>C	Polymorphism	–	–	–	–	LVNC (Tang et al., 2010)	
m.3397A>G	Reported	Benign	Likely polymorphism	Pathogenic	Benign	LVNC (Arbustini et al., 2016); ADPD (Cavelier et al., 2001)	
m.4316A>G	Reported	Polymorphic	Pathogenic	Uncertain significance	Benign	HCM with hearing loss (Chamkha et al., 2011)	
m.6340C>T	Reported	Benign	Likely polymorphism	–	Benign	KSS (Saldaña-Martínez et al., 2019); Prostate cancer (Petros et al., 2005)	
m.9025G>A	Reported	Likely pathogenic	Pathogenic	–	Benign	CMP (Kargaran et al., 2020); motor neuropathy, LS, colon cancer (López-Gallardo et al., 2014)	
m.9214A>C	Polymorphism	Likely pathogenic	Pathogenic	–	–	CMP (Alila-Fersi et al., 2017; Kobayashi et al., 2011)	
m.9804G>A	Reported	Benign	Polymorphism	Conflicting interpretations of pathogenicity	Conflicting interpretations of pathogenicity	LHON (Lazdinyte et al., 2019); keratoconus (Xu et al., 2021); TOF (Tansel, Pacal & Ustek, 2014; MELAS (Wani et al., 2007); OAG (Inagaki et al., 2006)	
m.9856T>C	Reported	Likely benign	Likely polymorphism	–	Likely benign	LVNC (Liu et al., 2013); gout (Tseng et al., 2018)	
m.10237T>C	Reported	Likely pathogenic	Pathogenic	Pathogenic	Benign	LHON (Yu-Wai-Man & Chinnery, 2021)	
m.11084A>G	Conflicting reports	Likely pathogenic	Likely pathogenic	Pathogenic	Benign	ADPD (Takasaki, 2008); MELAS (Lertrit et al., 1992)	
m.11087T>C	Polymorphism	Likely pathogenic	Pathogenic	–	Benign	LHON (Meng et al., 2014); T2D (Jiang et al., 2021)	
m.11361T>C	Polymorphism	Pathogenic	Pathogenic	–	Benign	Neuroblastoma (Calabrese et al., 2016)	
m.11778G>A	Confirmed	Pathogenic	Pathogenic	Pathogenic	Pathogenic	LHON (Poincenot, Pearson & Karanjia, 2020; progressive dystonia (Berardo et al., 2020)	
m.14420C>T	Polymorphism	Pathogenic	Pathogenic	–	Likely benign	LHON (Korkiamaki et al., 2013)	
m.14894T>C	Reported	Pathogenic	Pathogenic	–	–	LHON (Emmanuele et al., 2013)	
Notes:

ADPD, Alzheimer’s disease and Parkinsons’s disease; CMP, cardiomyopathy; HCM, hypertrophic cardiomyopathy; KSS, Kearns Sayre syndrome; LHON, Leber’s hereditary optic neuropathy; LS, Leigh’s syndrome; LVNC, left ventricular non-compaction cardiomyopathy; MELAS, Mitochondrial myopathy, encephalopathy, lactic acidosis, and stroke; OAG, open-angle glaucoma; T2D, type 2 diabetes; TOF, Tetralogy of Fallot. Variant(s) in bold is confirmed pathogenic.

According to MITOMAP:

Reported, ≥1 publications have considered the mutation as possibly pathologic.

Confirmed, ≥2 independent laboratories reported on the pathogenicity of a specific mutation, generally being pathogenic.

Haplogroup analysis

The CMP patients were assigned to different haplogroups according to their whole mitochondrial genome sequence. The haplogroup distribution pattern was similar between DCM and HCM, with haplogroup M as the most prevalent haplogroup among the Malaysian CMP patient cohort. For DCM, patients with haplogroup M accounted for 41.4%, followed by 21.8% of haplogroup B, 16.1% of haplogroup F, 9.2% of haplogroup R, 3.4% of haplogroup N, 2.3% of both haplogroups D and E, 1.1% each for haplogroups C, H and Y, and none for haplogroups G, I, Q and U.

As for HCM, the haplogroup with the highest frequency was haplogroup M (29.3%), followed by 20.7% of haplogroup B, 12.1% for both haplogroups F and R, 8.6% of haplogroup D, 3.4% each for haplogroups E, G and U, 1.7% each for haplogroups C, I, N and Q, and none for haplogroups H and Y (Table 5). Fisher’s exact test showed no significant difference between CMP subtypes and haplogroups (P values ≥ 0.05).

Table 5 Distribution of mitochondrial haplogroups in Malaysian CMP patients.

Haplogroup	Frequency, n (%)	P value	OR (95% CI)	
DCM (N = 87)	HCM (N = 58)	
B	19 (21.8)	12 (20.7)	1.0000	1.0711 [0.4747–2.4169]	
C	1 (1.1)	1 (1.7)	1.0000	0.6628 [0.0406–10.8125]	
D	2 (2.3)	5 (8.6)	0.1161	0.2494 [0.0467–1.3320]	
E	2 (2.3)	2 (3.4)	1.0000	0.6588 [0.0902–4.8140]	
F	14 (16.1)	7 (12.1)	0.6319	1.3973 [0.5269–3.7054]	
G	0 (0.0)	2 (3.4)	0.1583	0 (N/A)	
H	1 (1.1)	0 (0.0)	1.0000	N/A (N/A)	
I	0 (0.0)	1 (1.7)	0.4000	0 (N/A)	
M	36 (41.4)	17 (29.3)	0.1613	1.7024 [0.8384–3.4569]	
N	3 (3.4)	1 (1.7)	0.6500	2.0357 [0.2065–20.0638]	
Q	0 (0.0)	1 (1.7)	0.4000	0 (N/A)	
R	8 (9.2)	7 (12.1)	0.5892	0.7378 [0.2521–2.1590]	
U	0 (0.0)	2 (3.4)	0.1583	0 (N/A)	
Y	1 (1.1)	0 (0.0)	1.0000	N/A (N/A)	
Note:

OR, odds ratio; CI, confidence interval; N/A, not available.

Discussion

In recent years, numerous mtDNA variants have been identified and reported to be associated with CMP (Govindaraj et al., 2014; Kargaran et al., 2020). However, only limited number of studies were carried out in Asia, and none was conducted in Malaysia. This is the first whole mitochondrial genome sequencing study to elucidate the role of mtDNA variants in Malaysian CMP patients. The interpretation of the pathogenicity of mtDNA variants can be challenging due to conflicting reports from different laboratories. Guidelines developed by The American College of Medical Genetics (ACMG) and the Association of Molecular Pathology (AMP) are essential to enhance consistency in classifications of variants (Richards et al., 2015). Recently, more precise and specific guidelines for mtDNA variants interpretation and specification of ACMG/AMP standards were developed (McCormick et al., 2020), suggesting the utilisation of comprehensive databases such as MITOMAP, HmtDB, HmtVar, MSeqDR and ClinVar for mtDNA variant assessment. In favour of concise result reporting, our study utilised the suggested databases during data analysis for variants detected in our CMP patient cohort.

In the present study, 18 novel variants were observed, of which m.8573G>C, m.11916T>A and m.11918T>G variants were predicted to be potentially pathogenic via in-silico analysis. These three variants were not reported in gnomAD. Variant m.8573G>A, which was at the same position as one of the novel variants (m.8573G>C) in this study, was reported to be benign in ClinVar (Ganetzky et al., 2019). However, the novel variant m.8573G>C found in this study was a transversion mutation, which has more a detrimental effect due to the change of ring structure in nucleic acid compared to transition mutation in m.8573G>A (Lyons & Lauring, 2017). Variant m.8573G>C caused a substitution of glycine to alanine at a highly conserved region (CI = 93.3%) in MT-ATP6. Although glycine and alanine are both non-polar, the replacement of glycine at the conserved region could alter the protein structure. Glycine is the only amino acid with hydrogen but not carbon as its side chain, making it to be extra versatile as it can reside in difficult part of protein, such as beta turn and loop in secondary structure (Betts & Russell, 2003). Five patients (three DCM and two HCM) in this study harboured heteroplasmic m.8573G>C variant, with loads ranging from 6.75% to 12.66%. Most disease-associated pathogenic mtDNA variants tend to exist as heteroplasmic (Ye et al., 2014). Interestingly, four patients had the same haplogroup, which was B5a1, whereas one patient had a different haplogroup, which was M8a2, suggesting the pathogenicity of this variant regardless of haplogroup background.

On the other hand, a DCM patient was observed to harbour m.11916T>A variant, which caused a change from phenylalanine to tyrosine at 93.3% conserved region in MT-ND4. Although phenylalanine and tyrosine are both aromatic and non-polar, tyrosine has an extra –OH in the phenyl ring which would enhance the solubility. The change in hydrophilicity would alter the structure of the protein. Another variant from MT-ND4, m.11918T>G was observed in a DCM patient. This variant caused a change of serine to alanine. Serine is a small, polar amino acid that often acts as an active site due to the presence of R-group. In contrast alanine is non-polar, hydrophobic and with non-reactive side chain. Although the alteration occurred at a region conserved only in mammals and insect (CI = 80.0%), the substitution of amino acid with completely different properties could impact the folding and function of protein, leading to the malfunction of Complex I and reduced energy production that caused CMP.

According to McCormick et al. (2020), a variant with allele frequency <0.00002 fulfils ACMG pathogenicity code PM2 and coding variant that is predicted to have a deleterious effect by APOGEE (score >0.5 = pathogenic) fulfils ACMG pathogenicity code PP3. All the three variants mentioned above fulfilled the criteria for PM2 and PP3. At present, these variants could only be classified as uncertain significance because they did not meet the criteria for likely pathogenic and pathogenic classification. However, in view of remarkable results in the in-silico analysis and the fact that these variants are located in highly conserved regions, functional studies to elucidate the structural and biochemical effect of these variants are needed to denote their negative consequences to the mitochondrial function.

This study also revealed 17 variants that were associated with CMP and/or mitochondrial diseases. Among the 17 variants, two variants, m.1555A>G and m.11778G>A, were reported with confirmed pathogenicity. In this study, there was one HCM patient with haplogroup E1a2 who was observed to harbour the homoplasmic m.1555A>G variant. This variant was frequently reported in sensorineural hearing loss (SNHL) (Jiang et al., 2015; Mutai et al., 2017). Besides, this variant also altered the secondary protein structure of 12S ribosomal RNA, creating a binding site for aminoglycoside and causing ototoxicity (Prezant et al., 1993). Furthermore, the rate of protein synthesis was also disrupted due to the alteration in protein conformation and translation (Hobbie et al., 2008). Biochemical test also showed that cell harbouring this variant had decreased ATP production and increased ROS production (Wrzesniok et al., 2013). All the biochemical defects above could impact the protein synthesis needed for OXPHOS in mitochondria and lead to energy deficiency in cardiomyocytes. This variant was reported heteroplasmic in different type of tissues in a woman with maternally inherited CMP (Santorelli et al., 1999). Biochemical studies from homogenates of muscle and skin also revealed a decrease in complex activities in the respiratory chain. Furthermore, the m.1555A>G variant has been associated with a paediatric patient with SNHL and DCM (Skou et al., 2014). On the other hand, another paediatric patient with DCM was also reported to harbour the m.1555A>G variant in homoplasmic state. The proband’s sister and brother died at their infancies due to DCM as well (Alila-Fersi et al., 2017).

Another pathogenic variant with confirmed status observed in this study was m.11778G>A, which was reported in 90% of Leber’s hereditary optic neuropathy (LHON) patients of Asian descent (Jia et al., 2006). LHON is an ophthalmologic disease that also often manifests systemic abnormalities such as CMP (Finsterer & Zarrouk-Mahjoub, 2016). This missense variant resulted in the substitution of arginine to histidine, which share the same properties as a basic polar amino acid. However, the change occurred in a 100% conserved region of MT-ND4; therefore, the impact should not be underestimated.

Cybrid studies had revealed that m.11778G>A caused reduction in maximal respiration rate and Complex I activity (Brown et al., 2000). Besides, cybrids with m.11778G>A also showed increased ROS levels and apoptosis rate (Giordano et al., 2011). A study from Japan reported that a woman with LHON and CMP harboured m.11778G>A (Watanabe, Odaka & Hirata, 2009). Strikingly, the DCM patient in our study who harboured the m.11778G>A variant was also found to carry the only heteroplasmic variant, m.9804G>A, among the 17 disease-associated variants. Although the pathogenicity of m.9804G>A was uncertain due to conflicting evidence, in-silico analysis from PROVEAN, MutPred, Mitoclass.1, CADD, PhD-SNP and APOGEE predicted it to be pathogenic and have deleterious effect. Furthermore, m.9804G>A is located in a highly conserved region (CI = 93.3%) of MT-CO3. It was plausible that both variants can act synergically to cause malfunction of Complex I (MT-ND4) and IV (MT-CO3) and disrupt the ETC, which then reduced the ATP production and elevated the ROS level. Energy deficiency and apoptosis are detrimental to cardiomyocytes and may result in CMP.

Among the 17 variants, the only tRNA variant detected in a HCM patient in the study, m.4316A>G, was predicted to be possibly benign by MitoTIP. However, it is located at A58 of the T-loop in tRNAIle encoded by MT-TI, which is also highly conserved among 15 species (CI = 93.3%). According to the Mamit-tRNA database, the adenine is conserved in 31 out of 34 Euarchontoglires organisms. Besides, m.4316A>G was reported in a patient with HCM and profound hearing loss (Chamkha et al., 2011). It is also located next to m.4317A>G, which was first described in fatal infantile CMP (Tanaka et al., 1990), followed by multiple mitochondrial diseases such as deafness and myopathy (Zheng et al., 2020). Moreover, variant m.4317A>G was depicted to alter the structure by rearrangement of T-arm in tRNAIle. Study has shown the alternation affects the aminoacylation and thus causing impaired tRNA conformation and stability (Meng et al., 2018). The same study also demonstrated the synergy effect of m.4317A>G with m.1555A>G in deafness, whereby the cell lines carrying both variants exhibited significant deterioration of mitochondrial function than cell lines carrying only m.1555A>G variant (Meng et al., 2018). All these outcomes suggest the potential pathogenicity of m.4316A>G in CMP.

The m.9025G>A variant identified in this study is localised at a conserved region among 15 species with CI = 100%. The variant caused the change of non-polar glycine to polar serine at position 167. The alteration also introduced changes in the interaction between neighbouring residues whereby serine forms polar contacts with isoleucine at 164 and methionine at 171. On top of that, glycine at position 167 plays its role as a residue in the trans-membrane functional domain of ATP synthase membrane subunit 6 protein. The variant was detected in members of a family with HCM (Kargaran et al., 2020). Besides CMP, the variant was also reported in patients with mitochondrial disorder such as Leigh syndrome, motor neuropathy and 3-methylglutaconic aciduria type IV (López-Gallardo et al., 2014). The significance of m.9025G>A variant in pathogenesis of CMP was further supported by a functional study which observed a decreased baseline respiration and increased ROS production, thereby promoting apoptosis (Ganetzky et al., 2019). Besides, m.9025G>A was predicted to be ‘deleterious’, ‘pathogenic’ and ‘damaging’ by 11 in-silico predictors in this study. All these findings suggest that the change of amino acid with different properties by m.9025G>A would impair the active binding site of a protein and hinder the OXPHOS that is responsible for ATP production, followed by insufficient energy supply to the cardiomyocytes and consequently leading to CMP (Heidari et al., 2020).

Despite observing novel and CMP and/or mitochondrial disease-associated variants among the Malaysian CMP patients, several limitations in this study warrant careful interpretation of the findings. First, the control cohort was not included in the study. Healthy age and ethnically matched controls are vital as a comparison group to exclude the founder variant in certain populations during the analysis. Second, the mitochondrial genome spectrum in family members of study subjects was not evaluated; hence, screening of segregation and penetrance of variant cannot be performed. Third, genetic analysis of nuclear-encoded genes such as sarcomeric genes was absent, thus the potential effect of nuclear mutation among the patients is not known.

Conclusion

In summary, the whole mitochondrial genome sequencing of CMP patients in Malaysia revealed a total of 1,077 variants. After excluding ancestral variants, haplogroup markers and variants located in the control region, there were 512 variants remaining for further analysis. Amongst these variants, 18 were novel and 17 were CMPs and/or mitochondrial disease-associated variants. Three novel variants (m.8573G>C, m.11916T>A and m.11918T>G) were potentially pathogenic, as in-silico analysis predicted them to have deleterious effects. Two confirmed pathogenic variants (m.1555A>G and m.11778G>A) were also observed among the CMP patients in this study. Although the results are preliminary and the functional significance of the variants is yet to be elucidated, the findings of this study can provide fundamental insight into the distribution of mitochondrial genome variants in Malaysian CMP patients.

Supplemental Information

Supplemental Information 1 Supplementary Tables.

Click here for additional data file.

Supplemental Information 2 Raw data: Demographic, echocardiographic, mitochondrial haplogroup.

Click here for additional data file.

We would like to thank all of the participants involved in this study.

Additional Information and Declarations

Competing Interests

Author Contributions

Human Ethics

Data Availability

The authors declare that they have no competing interests.

Sheh Wen Kuan performed the experiments, analyzed the data, prepared figures and/or tables, authored or reviewed drafts of the paper, and approved the final draft.

Kek Heng Chua conceived and designed the experiments, authored or reviewed drafts of the paper, and approved the final draft.

E-Wei Tan performed the experiments, authored or reviewed drafts of the paper, and approved the final draft.

Lay Koon Tan conceived and designed the experiments, authored or reviewed drafts of the paper, and approved the final draft.

Alexander Loch conceived and designed the experiments, authored or reviewed drafts of the paper, and approved the final draft.

Boon Pin Kee conceived and designed the experiments, analyzed the data, authored or reviewed drafts of the paper, and approved the final draft.

The following information was supplied relating to ethical approvals (i.e., approving body and any reference numbers):

This study was approved by the Medical Ethics Committee of the University of Malaya Medical Centre (MECID NO.: 20152-1016) and the Ethics Committee of the National Heart Institute of Malaysia [IJNEC NO.: IJNEC/08/2015(3)].

The following information was supplied regarding data availability:

The sequencing raw data generated in this study is available at the European Nucleotide Archive (ENA): PRJEB43465.

The demographic, echocardiographic and mitochondrial haplogroup data are available in the Supplemental File.

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
