# Peer review of "Whole mitochondrial genome sequencing of Malaysian patients with cardiomyopathy"

_PeerJ, doi:10.7717/peerj.13265_

## Round 0.1 · original submission · Major Revisions

Reviewers have commented against the acceptance of the manuscript in its current form. The manuscript suffers from serious concerns regarding the implemented protocol as well as the presentation of the data. Please revise in light of reviewers' comments and resubmit accordingly.

Reviewer 1 ·

Basic reporting

English should be improved
• Examples:
• The findings of this study are crucial to shed light on the distribution of mitochondrial mutations in the Malaysian CMP patients
• HCM is characterised by left ventricular hypertrophy due 48 to cardiomyocyte hypertrophy and DCM is characterised by enlarged cardiac chambers and 49
The exact aetiology of CMP remains 53 inconclusive. It is generally accepted that environmental and genetic factors play a vital role in the 54 onset of CMP
385 Besides, it was also not rare to find reported defects and deficiency in Complex III, especially 386 MT-CYB (Hagen et al. 2013a) and Complex IV
Genetic defect in MT-tRNA and MT-rRNA adversely affect the transcription, translation and synthesis of mitochondrial protein…….
Comment2
The figure :Heatmap of conservation index (CI) and pathogenicity of variants predicted by in-silico tools should be more clear and readable

Experimental design

Comment1
You restate that the identified variants are pathogenic. I would, however, state that you have not undertaken mitochondrial genetics studies in their mother and siblings, and. In studied patients parents were examined??, but there were no explanations/data about parents. More information about the genotype of parents should be added
« except m.9804G>A that was heteroplasmic with 21.4% load (sporadic inherited ??)
Comment2
There was no description of a control group for MC sample in Methods and Materials.

Validity of the findings

no comment

Additional comments

no comment

Reviewer 2 ·

Basic reporting

The introduction gives sufficient background into the study. In the methods, it is unclear as to whether these patients have had the nuclear encoded genes associated with cardiomyopathy sequenced such as sarcomeric gene - if this has or hasn't occurred it should be clearly indicated.
The results section could be substantially modified to make it more clear and concise as only 207 of the 1077 need any discussion, I would re-mention Figure 1 indicating here where the variants were filtered out in the text and why, for example 'we performed a multi-step filtering procedure as illustrated in Figure 1. In the first step X variants with a global allele frequency of X% were removed etc'. The three novel synonymous variants do not need mentioning either. The section on haplotypes could also be shortened especially with the inclusion of this data within the figure 1 filtering protocol.
Table 3 and 4 contain too much unnecessary information, at least 13 of the variants presented are haplotype-related or clearly polymorphisms.
Discussion sections of this manuscript could be made more concise.

Experimental design

The analysis of novel variants for pathogenicity lacked rigour. The selection of species included within the conservation index for evaluation of protein coding genes was not appropriate. All species included were primates, for this analysis to be performed correctly a selection of at least 10 diverse species from humans, primates, marsupials, mammals, birds, fish, insects and yeast should be used. A CI score of 10 or greater is considered highly conserved, and 8 or greater conserved, any less than 8 is not considered conserved. For analysis of tRNA variants, the structural significance of position, base pairing and the conservation of the nucleotide should be investigated using MitoMaster, MitoTIP and Mamit-tRNA (http://mamit-trna.u-strasbg.fr/).

In the discussion, there was no mention of the alternate change at m.8573 which is reported in ClinVar as benign. The Serine residue disrupted by the m.11918T>G variant is only conserved down to mammals and the conservation index calculated in this paper artificially inflates the importance of the serine. All CI's of novel variants need to be recalculated.

Validity of the findings

The conclusions about complex I being the most common complex which possessed disease-associated variants is untrue, firstly these variants have not been shown to be disease-associated and is most likely due to the fact that complex I contains the most number of mitochondrial encoded subunits and chance.
The authors should take into consideration the levels of heteroplasmy/homoplasmy in tested tissues can also be vastly different to affected tissues, some variants such as the m.11778G>A variant can also have decreased heteroplasmy in blood with age and also show incomplete penetrance within families.

---

## Round 0.2 · accepted · Accept

The manuscript is significantly improved by the authors and can be accepted in its current form.